# Yeast Oral Delivery of DAF16 shRNAs Results in Effective Gene Silencing in *C. elegans*

**DOI:** 10.3390/cimb47070570

**Published:** 2025-07-20

**Authors:** Benedetta Caraba, Arianna Montanari, Emily Schifano, Fabiana Stocchi, Giovanna Costanzo, Daniela Uccelletti, Cristina Mazzoni

**Affiliations:** 1Department of Biology and Biotechnologies “C. Darwin”, Sapienza University of Rome, 00185 Rome, Italy; benedetta.caraba@uniroma1.it (B.C.); ari.montanari@uniroma1.it (A.M.); emily.schifano@uniroma1.it (E.S.); stocchi.1841666@studenti.uniroma1.it (F.S.); daniela.uccelletti@uniroma1.it (D.U.); 2Institute of Molecular Biology and Pathology, National Research Council (IBPM-CNR), 00185 Rome, Italy; giovannamaria.costanzo@cnr.it; 3Water Research Institute, National Research Council (IRSA-CNR), 00010 Rome, Italy

**Keywords:** yeast, oral delivery, nematodes, RNA interference, Biocontrol Agent

## Abstract

Plant Parasitic Nematodes (PPNs) are a major problem in agriculture. Damage caused by PPNs has been estimated at USD 80–157 billion annually. The estimates could be even worse in the future in the context of a growing world population in a climate change scenario and with the removal/reduction in the use of some nematodicides due to the strong ecological impact. Biocontrol Agents (BCAs) currently constitute only 8.8% of the general pesticide market. With a view to an ecological transition, the transition from pesticides to biopesticides represents an important challenge that appears necessary not only for organic farming, but also in so-called integrated agriculture. Among the possible BCAs, microorganisms, and in particular yeast, which enjoys the GRAS (Generally Recognized As Safe) status, have the advantage of being able to be produced on a large scale by fermentation on waste substrates at low cost. In this paper, as proof of concept we constructed yeast strains expressing short hairpin RNAs (shRNAs) targeting the *daf-16* gene in *C. elegans*. We demonstrate that oral ingestion of yeast cells expressing DAF16 shRNA is efficient in lowering *daf-16* expression and lifespan, suggesting a sustainable RNA interference-based strategy to inhibit the development of PPNs.

## 1. Introduction

RNA interference (RNAi) has emerged as a promising biotechnological tool in agriculture for the targeted control of crop pests. This endogenous cellular mechanism relies on the sequence-specific degradation of messenger RNA (mRNA) mediated by double-stranded RNA (dsRNA), resulting in the suppression of gene expression and the inhibition of corresponding protein synthesis [1,2].

When ingested or taken up by pests, dsRNA biopesticides trigger RNA interference (RNAi) against specific pest mRNAs, leading to reduced growth or mortality [3,4,5,6,7]. These biopesticides, designed with sequences targeting essential proteins, have been effectively applied against a wide range of pests, including insects, nematodes, viruses, and fungi [3,8,9,10,11]. Two primary strategies for dsRNA delivery have been developed—either through genetically modified (GM) crops that express dsRNA biopesticide within plant tissues [3,4,12] or via exogenous application of dsRNA onto plant surfaces [11,13]—both enabling pests to ingest the dsRNA during feeding and thereby triggering the RNAi response. While effective, these approaches are associated with notable limitations, including ecological concerns related to off-target effects and substantial production costs [14,15,16,17,18,19].

To overcome these challenges, we propose the yeast *Saccharomyces cerevisiae* as a low-cost biological device for the delivery of RNAi to protect crops from pests. *S. cerevisiae* is a Generally Recognized As Safe (GRAS) organism, has a long-standing history in food and fermentation industries, and is increasingly being explored for applications in sustainable agriculture, including its role as a biofertilizer and biopesticide [20]. Despite the high degree of conservation of the main eukaryotic regulatory processes, some of the fundamental genes for interfering RNA (Dicer and Argonaute) are not present in yeast and this mechanism of gene expression regulation has not been observed [21]. The absence of Dicer and Argonaute causes bioengineered dsRNAs to accumulate in yeasts, making it an ideal system to produce shRNAs. The use of yeast as a means of expression and delivery of interfering RNA through the oral administration has been reported both in mouse and insect models. In the first case, the yeast vector carries the shRNA expression cassette under the control of a mammalian U6 promoter followed by a short leader sequence of nuclear RNA. In this case, yeast cells were administered orally to mice which acquired them from the dendritic cells of the intestine, and the vector was transcribed and processed by the host cells [22,23,24]. In insect models, genetically modified yeast expressing dsRNA targeting *Drosophila suzukii*γ-tubulin reduced larval survival [25]. Similarly, an oral delivery *S. cerevisiae* strain engineered to produce shRNAs against the *Aedes aegypti* orthologs of fasciculation and elongation protein zeta 2 (fez2) and leukocyte receptor cluster (lrc) member resulted in silenced target gene expression and a significant increase in larval mortality [26].

In the context of plant–nematode interactions, nematode infestation triggers the oxidative burst in plants, resulting in the accumulation of reactive oxygen species (ROS) in the apoplast. In response, nematodes activate antioxidant pathways, with the Forkhead box O (FoxO) transcription factor DAF-16 and the transcription factor SKN-1 serving as a central regulator, which induces the activation of the antioxidant pathways, resulting in a robust cellular response that renders nematodes extremely resistant to stress. [27]. The *daf-16* gene encodes three main isoforms, each controlled by a distinct promoter and expressed in specific tissues, producing proteins that differ at their N-terminal regions. Among them, *daf-16b* is the most divergent as its expression is largely restricted to neurons, the pharynx, and the somatic gonad, whereas *daf-16a* and *daf-16f* are more broadly distributed, playing key roles in regulating lifespan, dauer formation, and stress resistance, with minimal contribution from *daf16b*. In contrast, *daf-16b* is primarily involved in neuronal morphology and behavior. Despite these distinct functions, some overlap exists. For example, both *daf-16a* and *daf-16b* contribute to innate immunity in adults, although *daf-16a* has a more dominant role [28]. Furthermore, all three isoforms are involved in suppressing the expression of an adult cell fate marker in the hypodermis during the dauer stage. Given its pivotal role in stress response, *daf-16* represents an ideal target to sensitize nematodes during infection and make them more susceptible to the natural defenses of plants [29]. Previous work has demonstrated that transgenic *A. thaliana* and *N. tabacum* plants expressing hairpin-derived dsRNA targeting *MiDaf16-like1* and *MiSkn1-like1* genes, orthologous to *C. elegans*
*daf-16* and *skn-1* in the plant-parasitic root-knot nematode *Meloidogyne incognita*, resulted in partial resistance to infection, indicating that these two transcription factors are potential targets for the development of biotechnological tools for nematode control [30].

As a proof of concept, we engineered *S. cerevisiae* strains to express shRNAs targeting *C. elegans*
*daf-16*. Nematodes fed with these yeast strains exhibited significant reductions in *daf-16* mRNA levels, indicating effective gene knockdown and a shorter lifespan.

Collectively, these findings support the feasibility of using yeast as a delivery system for RNAi to interfere with pest gene expression and enhance crop protection in a cost-effective, specific, and sustainable manner.

## 2. Materials and Methods

### 2.1. Strains, Growth Conditions

*S. cerevisiae* and *C. elegans* strains used in this work are described in Table 1.

Yeast cells were grown at 28 °C in YPD (1% yeast extract, 2% bacto-peptone, 2% glucose) and SD (0.67% yeast nitrogen base without aminoacids, 2% glucose) supplemented with auxotrophic requirements. Solid media were obtained by the addition of 2% Bactoagar. When needed, antibiotics G418 (Sigma-Aldrich, G8168, St. Louis, MO, USA) and nourseothricin (Jena Bioscience, Jena, Germany, AB-102L) were added to YPD plates at the concentration of 200 mg/L and 100 mg/L, respectively.

Yeast cells were resuspended in M9 media (42 mM Na_2_HPO_4_, 24 mM KH_2_PO_4_, 9 mM NaCl, 19 mM NH_4_Cl, 1 mM MgSO_4_, 0.1 mM CaCl_2_, 2.0% Glucose, 0.5 μg/mL Thiamin) and administrated to *C. elegans* at a concentration of 6.75 × 10^8^ cells/mL. The nematodes were fed fresh preparations of the same transformed cells for four days.

The *C. elegans* strains used were wild-type N2 and *zIs356 [daf-16p::daf-16a/b::GFP + rol-6 (su1006)]*. Nematodes were grown on nematode growth medium (NGM) and fed with different yeast cultures. Fertile N2 adults were placed to lay embryos for 8 h on 3.5 mm Petri dishes plated with different conditions and then sacrificed. When the progeny became fertile (t0), 80 worms per condition were transferred to new plates plated with fresh yeast cultures and monitored daily. A worm was considered dead when it did not respond to touch.

*E. coli* DH5a strain (fhuA2 lac(del)U169 phoA glnV44 Φ80′ lacZ(del)M15 gyrA96 recA1 relA1 endA1 thi-1 hsdR17) was grown in LB (Yeast extract 0.5%, tryptone 1%, NaCl 1%). For plasmid selection, 100 mg/L Ampicillin was added.

### 2.2. Yeast Transformation

Yeast transformation was performed using the ONE-STEP protocol as previously reported [34]. Briefly, a colony grown on a YPD plate was resuspended in a reaction mixture containing 1 μg of plasmid DNA, 200 μL of PEG 60% (*w*/*v*), 60 μL of LiAc 1 M, 30 μL of DTT 1 M, and 5 μL of salmon sperm DNA that served as a DNA carrier. This transformation mix was incubated 30 min at 45 °C, plated on selective media with the appropriate auxotrophies or antibiotics, and grown at 30 °C for 3 to 4 days.

### 2.3. Plasmids and Recombinant Yeast Construction

Plasmids and primers used in this work are reported in Table 2 and Table 3, respectively.

For the construction of plasmids pshDaf16.1 and pshDaf16.2, the pairs of oligonucleotides shDaf16For1/Rev1 and shDaf16For2/Rev2 (0.5 mmol each) bearing *EcoRI* and *HindIII* protruding ends were annealed (30 s at 95 °C, ice for 1 min, 30 s at 95 °C, and 10 min at 56 °C) and ligated into the pYX232-mtGFP plasmid [35] digested by *EcoRI* and *HindIII* restriction enzymes. The digestion procedure opens the plasmid and removes the mtGFP insert from it. To obtain the empty vector pshDaf16.Ø, the pYX232-mtGFP plasmid digested with *EcoRI* and *HindIII* restriction enzymes was treated with Kleenow enzyme to the fill-in the overhanging sequences, ligated, and transformed into *E. coli* DH5a strain.

Genome editing was performed using the pCfB2312 and pCfB2311 plasmids described in [36], obtained through Addgene (Table 2). Plasmid DNA was extracted with an ISOLATE II Plasmid Mini Kit (Bioline, Meridian Bioscience, Memphis, TN, USA, BIO-52056).

Yeast cells were transformed with pCfB2312 plasmid, containing the Cas9 gene, according to [37], plated on YPD supplemented with 200 mg/L G418 sulfate, and incubated at 28 °C for three to four days. Resulting colonies were then double transformed with pCfB2311-plasmid and donor DNAs obtained by PCR amplification using pshDaf16.1 or pshDaf16.2 plasmids as templates with the primers sh232insFW/ACR3GFPinsREV primers listed in Table 3, and plated on YPD supplemented with both 200 mg/L G418 sulfate and 100 mg/L nourseothricin. pCfB2311 plasmids bear a gRNA which targets CAS9 onto the *ADE2* gene. The inactivation of the *ADE2* gene by the insertion of donor DNA determines the accumulation of a red pigment, allowing the screening of the recombinant strains on plates. After 4–6 days at 28 °C selected colonies were plated on YPD without antibiotics to allow the loss of plasmids. The screening of colonies was performed through PCR (primers listed in Table 3). PCR products were purified with the Gel/PCR DNA Fragments Extraction Kit (Geneaid, Biotech Ltd, New Taipei City, Taiwan, DF100) and sequencing was committed to Bio-Fab research s.r.l, Rome, Italy.

### 2.4. Yeast RNA Extraction and Primer Extension

For RNA extraction, cells corresponding to 4 OD_600_ were washed with H_2_O, resuspended in 200 μL of lysis buffer (0.5 M NaCl, 0.2 M Tris-HCl pH 7.5, 10 mM EDTA, 1% SDS), 200 μL of phenol-chlorophorm-isoamyl alcohol (PCI) 25:24:1 (Sigma, 77617), and ground by vortexing with micro-glass beads. After the addition of 300 μL of lysis buffer and 300 μL of PCI, cells were centrifugated at 10,000 rpm for 5′ at 4 °C, then the supernatant precipitated with 3 volumes of ethanol at −20 °C for 30′ (adapted from [38]). The precipitated nucleic acid was resuspended in 15 μL of RNase-free H_2_O. The integrity of RNA was tested via electrophoresis on agarose gel, 1% in TAE buffer 1 × (40 mM Tris, 20 mM Acetate and 1 mM EDTA) stained with Ethidium bromide. RNA was treated with DNaseI using a DNA-*free* TM kit (Invitrogen, Thermo Fisher Scientific, Waltham, MA, USA, AM1906).

Markers preparation was as follows: 100 pmol of oligonucleotides shDaf16Rev1 and shDaf16Rev2 were labeled with 30 μCi [γ-^32^P] ATP using 0.5 μL of polynucleotide kinase (EC 2.7.1.78, 10 U/μL, New England Biolabs, Ipswich, MA, USA; #M0201 L) in a total volume of 15 μL. This enzyme catalyzes the transfer and exchange of Pi from the γ-position of ATP to the 5′-OH terminus of polynucleotides, and the removal of the 3′-phosphoryl group from 3′-phosphoryl polynucleotides. One unit is defined as the amount of enzyme catalyzing the production of 1 nmol of phosphate to the 5′-OH end of an oligonucleotide from [γ-^32^P] ATP in 30 min at 37 °C. The volume was adjusted to 100 μL with bidistilled water and an equal volume of phenol–chlorophorm–isoamylic alcohol was added to stop the reaction. The samples were than precipitated by the addition of 3 μL of glycogen (Thermo Scientific, Thermo Fisher Scientific, Waltham, MA, USA, 20 μg/μL in water), 0.3 M (final volume) of sodium acetate pH 7.5, and 3 volumes of 96% ethanol, and incubated overnight at −20 °C. Samples were centrifuged at top speed for 20′, pellets were washed with 70% ethylic alcohol, and dehydrated (Savant SpeedVac Concentrator, Thermo Fisher Scientific, Waltham, MA, USA), 13,000 rpm, 10 min, room temperature, environmental atmospheric pressure). Pellets were resuspended in 20 μL formamide buffer (90% formamide, 10% bromophenol blue and xylene cyanol) and 5 μL were loaded onto 10% polyacrylamide gels (19:1 ratio acrylamide–N,N’-methylene-bis-acrylamide) containing 7 M urea in 1XTBE buffer.

The primer extension labeling reaction was as follows: 3 pmol of oligonucleotides Daf16.1_PriEx and Daf16.2_PriEx (Table 3) were labeled with 10 μCi of [γ-^32^P]ATP by polynucleotide kinase and treated as described above. This procedure provided a specific activity of 150,000.00 cpm/pmol.

The primer extension reaction was as follows: to determine the start site of transcription, 2 pmol of the specific labeled reverse primer (Daf16.1_PriEx or Daf16.2_PriEx) were added with 1 μg of purified RNA to the retrotranscription mixture of a SensiFAST cDNA Synthesis Kit (Meridian, Bioscience, Cincinnati, OH, USA, BIO-65053) kit, and the extension step was performed for 15′ at 48 °C to ensure the retrotranscription of highly structured RNAs. The inactivation step was performed at 85 °C for 5 min. Primer-extended fragments were purified using an equal volume of phenol–chloroform–isoamyl alcohol (25:24:1, Sigma-Aldrich, #77617), and then precipitated with 0.3 M Sodium Acetate, 10 µg glycogen (Thermo Scientific, #R0561), and three volumes of EtOH 96% at −20 °C for 30 min. Samples were then centrifuge at 13,000 rpm for 15 min, washed once with EtOH 70%, dehydrated (Savant SpeedVac Concentrator, 13,000 rpm, 10 min, room temperature, environmental atmospheric pressure), suspended in 5 µL of 100% formamide, loaded onto a 10% denaturing polyacrylamide gel containing 7 M urea along with the indicated markers, and run in 1XTBE buffer.

### 2.5. C. elegans Fluorescence Microscopy and Lifespan Analysis

Nematodes were grown on Nematode Growth Medium (NGM) and fed with 100 μL of 6.75 × 10^8^ cells/mL yeast cultures from embryos hatching.

For fluorescence microscopy at the stage of 1 day of adulthood, *daf-16*::GFP transgenic worms, fed with the indicated yeast strains from embryos hatching, were exposed at 37 °C for 1 h to induce a stress response. Then, worms were anesthetized with sodium azide (20 mmol L^−1^) (Sigma-Aldrich, St. Louis, MO, USA) and observed by a Zeiss Axiovert 25 microscope as described in [39]. The experiment was repeated three times and ten worms for every condition were used in each experiment. Images were taken at the time of exposure of 0.2 s [40]. The number of fluorescent nuclei was taken through Zeiss, Oberkochen, GermanyZEN Microscopy Software1.0 2011. For lifespan analysis, nematodes were grown in NGM and fed with 100 μL of transgenic shDaf16.Ø or shDaf16.2 cultures from the embryos hatching. Lifespan was performed at 16 °C and synchronous wild-type worms were transferred daily to new plates seeded with fresh lawns. They were scored as dead when they no longer responded to a gentle touch with a platinum wire. At least 80 nematodes per condition were used in each experiment. All lifespan assays were performed in triplicate.

### 2.6. Real Time-qPCR

The RNA of 200 1-day adults treated with different yeast preparations from embryos hatching was extracted using a miRNeasy Micro Kit (Qiagen, Venlo, Netherlands) and real time analysis with an I Cycler IQ Multicolor Real-Time Detection System (Biorad, Hercules, CA, USA). This was performed according to [41]. The selective primers (200 nM) for *daf-16* and *cdc-42* genes are reported in Table 3. Quantification was performed using a comparative Ct method (Ct = threshold cycle value). Briefly, the differences between the mean Ct value of each sample and the CT value of the housekeeping gene (*act-1*) were calculated as follows: ΔC_tsample_ = C_tsample_ − C*_act-1_*. Result was determined as 2^−ΔΔCt^, where ΔΔCt = ΔC_tsample_ − ΔC_tcontrol_. The experiment was performed in duplicate.

## 3. Results

### 3.1. Experimental Set up

The aim of this work was to construct a yeast strain expressing the specific shRNA for the nematode’s *daf-16* gene and demonstrate its effectiveness in vivo after the administration.

Two different shRNAs directed to the conserved region of *daf-16* were cloned into the yeast multicopy plasmid under the control of the pTPI constitutive promoter (Figure 1, step 1) and transformed into the CML39-11A yeast strain (Figure 1, step 2a). The expression cassettes pTPI-shRNAs-CYC1term were integrated into the yeast genome of the BY4741 strain by the CRISPR-CAS technique (Figure 1, step 2b). The two different engineered yeast strains, the plasmid expressing strain and the edited strain, were both administrated to the nematode model system (Figure 1, step 3). In order to quantify the interference on the target, *daf-16* expression and protein translocation into the nucleus were measured by real-time qPCR and fluorescence microscopy, respectively (Figure 1, step 4a and step 4b).

### 3.2. Design of shRNAs and Yeast Expression

Two shRNAs directed to sequences present in the conserved region of *daf-16* genes were designed (Table 3). The overall structure of the shRNAs comprises, from the 5′ end, 19 nucleotides of the passenger strand, 9 nucleotide of the loop and 19 nucleotides of the guide strand with an additional 5 U bases, which represents a widely used siRNA design already described in [42], forming the stem-loop structure described in Figure 2. The shDAF16.1 shows the canonical structure with a perfect pairing stem and a 9 nucleotides loop (Figure 2A), while the shDAF16.2 shows a deletion in the passenger strand between nucleotide 15 and 16, leading to an unpaired G nucleotide in position 31 (Figure 2B) which, in turn, promotes an alternative secondary structure where the 9 nucleotides of the loop (in green) form a dramatically smaller loop and a second bulge is present (Figure 2C).

Therefore, we checked the hypothetical secondary structure of these sequences by the informatic tool RNA structure (https://rna.urmc.rochester.edu/RNAstructure.html (accessed on on 5 January 2025) using the Fold algorithm that predicts the lowest free-energy structure and a set of low free-energy structures for a sequence, and it confirmed the second structure proposed for the shDAF16.2, with a probability even higher than the one proposed for shDAF16.1 (Figure 3).

This uncanonical secondary structure of shDAF16.2 could enhance the interference activity of the shRNA thanks to two different features: firstly, the guide sequence should be placed in the 3′ arm and begin with U, as this is the most preferred nucleotide for Ago association, therefore enhancing the preferential loading of the strand into RISC complex [43]; on the other hand, as described by Terasawa and colleagues [44], shRNAs presenting internal bulges due to deletions and four nucleotides loops, as in our case, have a significant gene-silencing activity.

To determine the transcription start site (TSS) of the shRNA constructs, we performed primer extension analysis, as shown in Appendix A. The labeled primers used for this assay are listed in Table 3. It is known that in yeast, RNA polymerase II (RNAPol II) typically initiates transcription at pyrimidine–purine (PyPu) dinucleotides, with a strong preference for a purine at the +1 position and a pyrimidine at the −1 position [45]. Consistent with this, both shDAF16 constructs exhibited TSSs mapping approximately 11 and 9 nucleotides upstream of the start of the shDAF16 sequence. Interestingly, for shDAF16.2, we observed a strong accumulation of a 38-nucleotide fragment. This could reflect either the use of an alternative downstream TSS or a possible impediment in reverse transcription caused by the secondary structure complexity of the shRNA molecule (Appendix A).

### 3.3. C. elegans shRNAs Administration and RNA Interference

The nematode model was used to analyze the interfering capability of the proposed yeast-based silencing system. The transgenic nematodes’ embryos, in which the GFP gene is fused to *daf-16*, were interfered by feeding with specific yeast strains. At the stage of 1 day of adulthood, the RNAi nematodes were exposed to 37 °C to induce the thermal stress response and subsequently were analyzed by fluorescence microscopy. Figure 4A shows the results obtained treating the nematodes with transgenic yeast cells expressing from the genome no RNAi (shDaf16.Ø), shDaf16.1, shDaf16.2, or a mix of the two cultures (shDaf16.1 + shDaf16.2). In Figure 4B, the same shDaf16 RNAi sequences were administered to the nematodes by using yeast cells transformed with plasmids (pshDaf16.Ø, pshDaf16.1, pshDaf16.2, or pshDaf16.1 + pshDaf16.2, respectively). As shown in the pictures, after exposing the RNAi nematodes to thermal stress, the DAF-16 protein is mainly localized inside the nucleus, appearing as dots into the worms’ bodies. Fluorescence appears to significantly decrease in nematodes treated with yeasts expressing shDaf16.2, compared to worms treated with yeasts expressing shDaf16.1 or control yeasts expressing no shRNA (shDaf16.Ø).

Notably, as shown in Figure 5, which reports the quantification of GFP-positive nuclei, nematodes treated with transgenic yeast expressing shDaf16.2 alone or in combination with strains expressing shDaf16.1 exhibited a significant reduction in the number of fluorescent nuclei compared to the shDaf16.Ø control group, by 30% and 40%, respectively. Almost similar results were obtained for the transformed yeast strains. In contrast, nematodes treated with both yeast strains expressing shDaf16.1 alone displayed a number of GFP-positive nuclei comparable to the control.

To evaluate the *daf-16* transcription level in different interfered nematodes, an RT-qPCR analysis was performed. Figure 6 shows a significant reduction of 70% and 80% in *daf-16* expression in nematodes, respectively, treated with transgenic yeasts expressing shDaf16.2 or a mix of both shDaf16.1 + shDaf16.2 sequences. When the interfering is performed by using transformed yeast cells, a 50% reduction in *daf-16* expression levels was observed after treatment with shDaf16.2 or shDaf16.1 + shDaf16.2. Treatments with the transgenic shDaf16.1 strain reduced the transcript levels of the *daf-16* gene by 20% as compared to control, whereas the same constructs in the transformed yeast cells did not alter the expression of the gene (Figure 6). An additional RT-qPCR was performed to detect the expression of another gene (e.g., *cdc-42*) to verify the specificity of the transgenic construct in targeting *daf-16*. The expression of *cdc-42* was unaffected in animals treated with shDaf16.2 compared to the control (Appendix A).

To evaluate the in vivo efficacy of the transgenic shDaf16.2 construct, a lifespan analysis was performed (Figure 7). Notably, the results revealed a significant reduction in the survival rate of animals treated with shDaf-16.2 compared to those fed with shDaf-16.∅. Specifically, by day 11 of adulthood, the survival rate of the shDaf-16.2 group was 55%, whereas the control group showed an 80% survival rate, compared to the initial population (100%).

## 4. Discussion

The yeast *Saccharomyces cerevisiae* has long been a point of reference for biotechnology due to its ease of cultivation, well-characterized genetics, and high amenability to genetic manipulation. Several studies have demonstrated its potential as a delivery system for recombinant nucleic acids, including RNA and short hairpin RNA (shRNA), to other living organisms [26,46,47]. Notably, *S. cerevisiae* lacks a functional RNA interference (RNAi) pathway, enabling the accumulation of bioengineered shRNAs within the yeast cells. Upon ingestion or interaction with a host organism, these shRNAs can then be processed by the host’s RNAi machinery, triggering gene silencing.

In this study, we demonstrated the feasibility of using recombinant yeast expressing short hairpin RNAs (shRNAs) to induce gene silencing in *Caenorhabditis elegans*. As a proof of concept, we engineered yeast strains producing two different shRNAs targeting the *C. elegans daf-16* gene, a key transcription factor involved in stress response and longevity [27].

Our results show that worms fed with yeast expressing the shDAF16.2 construct, which features a modified hairpin structure due to a nucleotide deletion, exhibited a marked reduction in the number of GFP-positive nuclei in *daf-16*-::GFP animals. Additionally, these worms displayed significantly lower *daf-16* mRNA levels compared to controls. These findings suggest that the altered shDAF16.2 construct may enhance the efficacy of RNAi, possibly by improving the stability or processing efficiency of the hairpin RNA within the host organism. In agreement with these observations, a lifespan reduction in animals fed with transgenic shDaf16.2 yeast was obtained.

The enhanced interference activity observed for shDAF16.2 may be attributed to its non-canonical secondary structure, which likely facilitates gene silencing through two complementary mechanisms. First, the guide strand resides in the 3′ arm and initiates with a uridine, a nucleotide known to favor binding to Argonaute proteins and thereby enhance RISC loading efficiency [43]. Second, the presence of internal bulges, along with a characteristic four-nucleotide loop observed in our construct, has been associated with significantly improved gene-silencing efficacy in shRNA molecules [44].

We also evaluated the efficiency of shRNA delivery by *Saccharomyces cerevisiae* through two different genetic strategies: expression from a multicopy plasmid and chromosomal integration of the shRNA expression cassette. Our results indicate that genome integration of the shDAF16 constructs was significantly more effective in silencing the *daf-16* gene in *C. elegans* compared to expression from multicopy plasmids.

We observed that the plasmid-based expression of shRNAs suffered from severe instability, with approximately 70% of yeast cells losing the plasmid even under selective conditions (-. This instability suggests that the expression of shRNAs may be toxic to yeast cells, at least in the strain used in this study. In contrast, yeast strains with the shRNA cassette integrated into the genome maintained stable expression and achieved efficient interference with *daf-16* expression, despite the lower gene dosage associated with a single genomic copy. These findings highlight that even low levels of shRNA expression, when stably maintained, are sufficient to trigger RNAi responses in the target organism.

Our data underlines the importance of the expression system used for shRNA delivery, favoring chromosomal integration over plasmid-based approaches in this context. Furthermore, strain selection emerges as a crucial factor for optimizing shRNA delivery platforms. Identifying or engineering yeast strains with improved tolerance to shRNA expression could further enhance the efficiency and stability of this RNAi-based strategy.

Taken together, these results provide valuable insights for the development of yeast-based systems for gene silencing applications, particularly in the context of biological control strategies targeting nematodes or other pests.

Importantly, the success of RNAi induction through dietary delivery of engineered yeast highlights a promising, low-cost, and scalable strategy for targeting pathogenic nematodes, particularly those that pose serious threats to agriculture. Current nematode control methods heavily rely on chemical nematicides, which are often associated with environmental concerns and the development of resistance [48]. Emerging strategies associated with integrated nematode management are currently under investigation [49], thus, an RNAi-based biocontrol approach using recombinant yeast could offer a sustainable and environmentally friendly alternative.

Future work will be necessary to assess the broader applicability of this method using multiple gene targets across different nematode species, including major agricultural pests. It will also be important to optimize the expression systems and shRNA designs to maximize gene silencing efficiency and minimize potential off-target effects. Overall, our study provides a foundation for the development of novel RNAi-based strategies to combat nematode infections in agricultural settings.

## Figures and Tables

**Figure 1 cimb-47-00570-f001:**
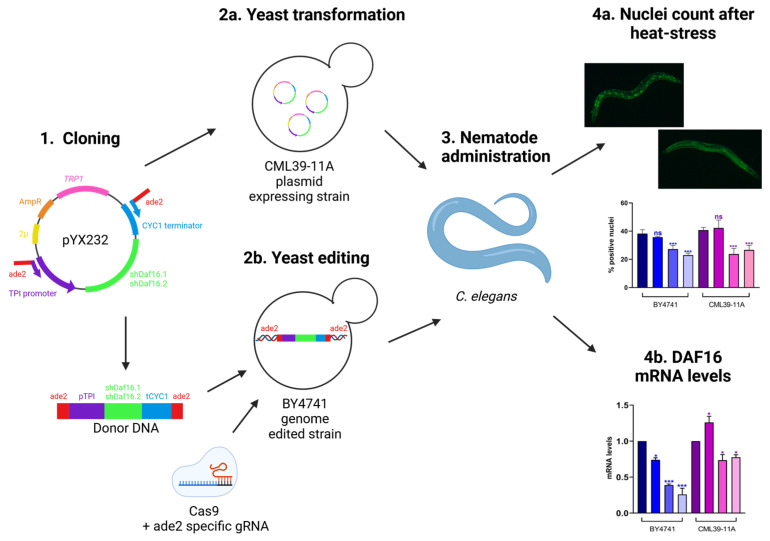
Experimental set up. Step 1: two different shRNAs directed to the conserved regions of *daf-16* mRNA were cloned in the pYX232 plasmid; Step 2a: the resulting plasmids were transformed in the CML39-11A strain; Step 2b: the resulting plasmids were used as a template for the production of the donor DNA used in genome editing of the BY4741 strain; Step 3: the resulting strains were administrated to nematode embryos; Step 4a: at day 1 of adulthood, nematodes were exposed at 37 °C for 1 h to induce the stress response and the translocation of DAF-16 in the nucleus and the percentage of fluorescence positive nuclei was analyzed; Step 4b: at day 1 of adulthood, RNA was extracted and *daf-16* mRNA levels were evaluated through real time qPCR (* *p* < 0.05, and *** *p* < 0.001; ns: not significant).

**Figure 2 cimb-47-00570-f002:**
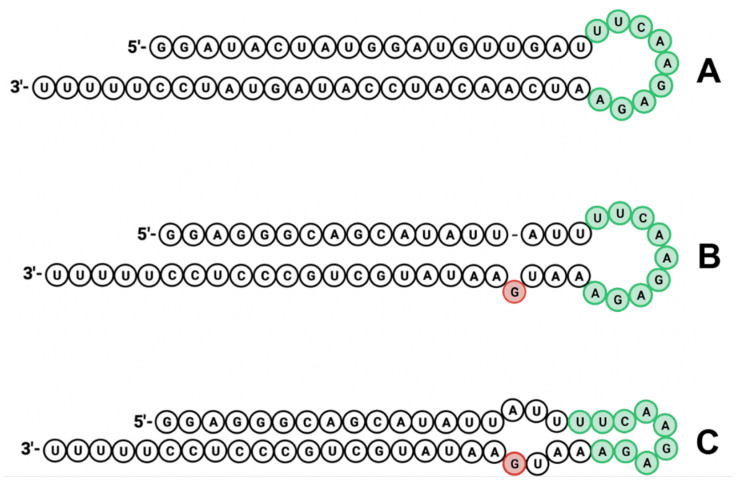
Secondary structures of the shRNAs. (**A**) Secondary structure of shDAF16.1, with perfectly paired stem and 9 nucleotides loop (in green). (**B**) Secondary structure of shDAF16.2, with the unpaired G nucleotide in position 31 (in red). (**C**) Alternative secondary structure of shDAF16.2, with the smaller loop and a second bulge.

**Figure 3 cimb-47-00570-f003:**
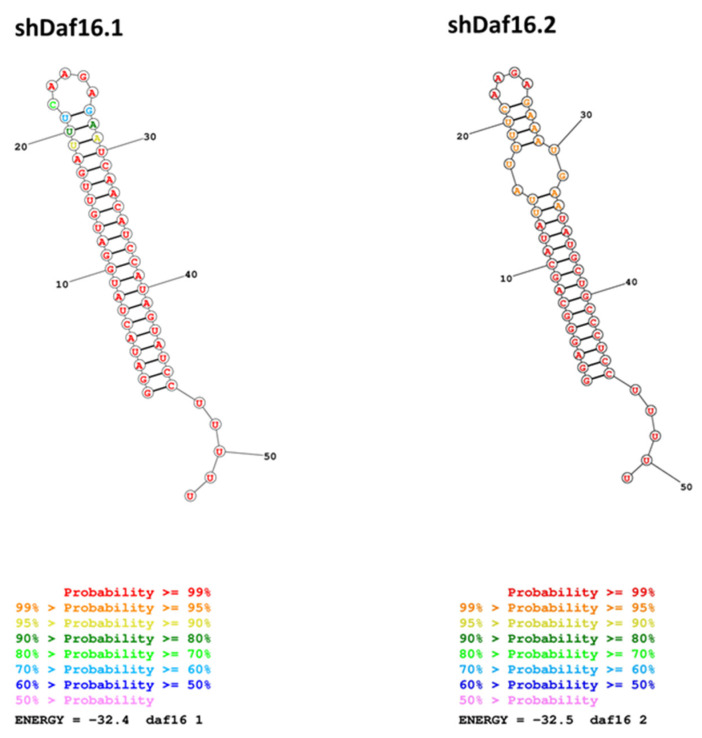
Hypothetical secondary structure of the two shRNAs calculated by RNAstructure (https://rna.urmc.rochester.edu/RNAstructure.html (accessed on 5 January 2025) using the Fold algorithm that suggests the lowest free-energy structure and a set of low free-energy structures for a sequence. The color of the bases represents the probability of the position of that base in the secondary structure.

**Figure 4 cimb-47-00570-f004:**
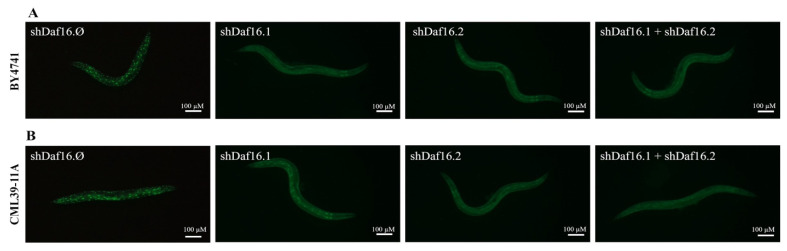
Fluorescence analysis of *daf-16*::GFP transgenic strain, at the stage of 1 day adult, treated with the different yeast preparations: transgenic BY4741 yeast cells in panel (**A**) and transformed CML39-11A yeast cells in panel (**B**). Data were obtained from three independent experiments (60 worms for each condition). Scale bar = 100 μm. Control: nematodes fed shDaf16.Ø.

**Figure 5 cimb-47-00570-f005:**
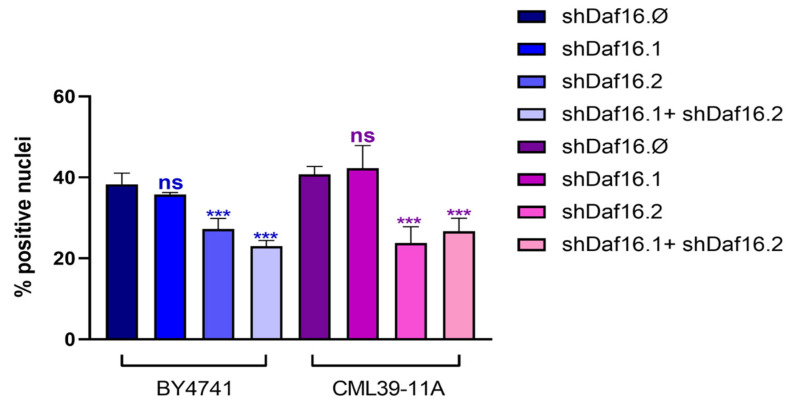
Percentage of GFP-positive nuclei counted in the *daf-16*::GFP transgenic strain at the 1-day adult stage, following treatment with strains expressing shRNAs, compared to yeast strains without shRNA expression, used as controls. Statistical analysis was evaluated by a one-way ANOVA with the Bonferroni post-test; asterisks indicate significant differences (*** *p* < 0.001; ns: not significant).

**Figure 6 cimb-47-00570-f006:**
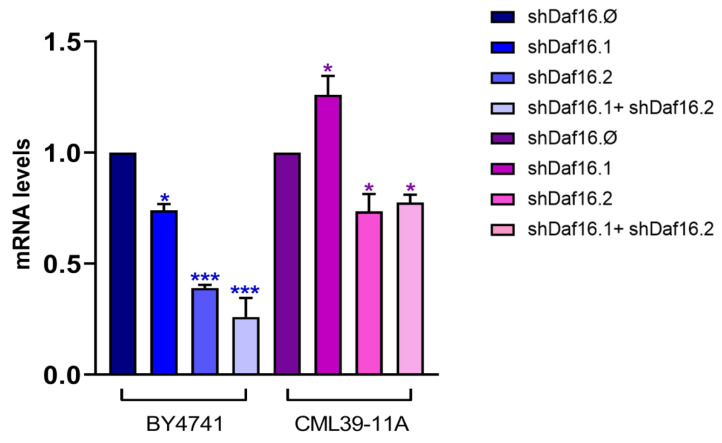
RT-qPCR analysis of *daf-16* transcript levels in 1-day adult wild-type worms treated with the different shRNAs from embryo hatching. Statistical analysis was evaluated by a one-way ANOVA with the Bonferroni post-test; asterisks indicate significant differences (* *p* < 0.05, and *** *p* < 0.001).

**Figure 7 cimb-47-00570-f007:**
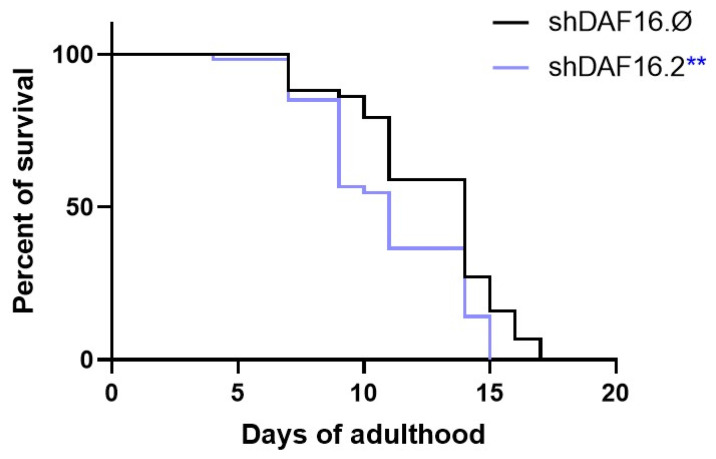
Lifespan experiment with wild-type N2 worms. A Kaplan–Meier survival plot of N2 worms fed from embryos hatching with transgenic shDaf16.Ø, taken as control, or shDaf16.2. *n* = 80 per data point in individual experiments. (** *p* < 0.01). The experiment was performed in triplicate.

**Table 1 cimb-47-00570-t001:** Yeast, bacterial, and worm strains used in this work.

*S. cerevisiae* Strains	Genotype	Reference
BY4741	Mat a, his3-Δ1, leu2-Δ0, met15-Δ0, ura3-Δ0	[31]
CML39-11A	MATa, ade1-101, his3-Δ1, leu2, ura3, trp1-289	[32]
BY4741 shDaf16.Ø	Mat a, his3-Δ1, leu2-Δ0, met15-Δ0, ura3-Δ0, ade2::shDaf16.Ø	This work
BY4741 shDaf16.1	Mat a, his3-Δ1, leu2-Δ0, met15-Δ0, ura3-Δ0, ade2::shDaf16.1	This work
BY4741 shDaf16.2	Mat a, his3-Δ1, leu2-Δ0, met15-Δ0, ura3-Δ0, ade2::shDaf16.2	This work
CML3911A pYX232-shDaf16.Ø	MATa, ade1-101, his3-Δ1, leu2, ura3, trp1-289, pYX232- shDaf16.Ø	This work
CML3911A pYX232-shDaf16.1	MATa, ade1-101, his3-Δ1, leu2, ura3, trp1-289, pYX232- shDaf16.1	This work
CML3911A pYX232-shDaf16.2	MATa, ade1-101, his3-Δ1, leu2, ura3, trp1-289, pYX232- shDaf16.2	This work
* **E. coli** * ** Strain**	**Genotype**	**Reference**
DH5-α	fhuA2 lac(del)U169 phoA glnV44 Φ80′ lacZ(del)M15 gyrA96 recA1 relA1 endA1 thi-1 hsdR17	[33]
* **C. elegans** * ** Strain**	**Genotype**	**Reference**
N2	*Caenorhabditis elegans* wild isolate	https://wormbase.org
TJ356	zIs356 [*daf-16p::daf-16a/b::GFP* + rol-6 (su1006)]	https://wormbase.org

**Table 2 cimb-47-00570-t002:** Plasmids used in this work.

Plasmid	Main Features	Reference
pYX232-mtGFP	2μ expression vector–Promoter: TPI–Selection marker: *TRP1*	[35]
pCfB2311	2μ expression vector–SNR52-gRNA-ADE2–Selection marker: Nat	[36]
pCfB2312	Cen expression vector–Tef1-Cas9–Selection: kanMX4	[36]
pYX232-shDaf16.Ø	2μ TPI-shDaf16.Ø (empty vector)	This work
pYX232-shDaf16.1	2μ TPI-ShDaf16.1	This work
pYX232-shDaf16.2	2μ TPI-ShDaf16.2	This work

**Table 3 cimb-47-00570-t003:** Oligonucleotides used in this work.

Primers	Sequence	Purpose
shDaf16For1	5′AATTCCCGGATACTATGGATGTTGA TTTCAAGAGAATCAACATCCATAGTATCCTTTTTA3′	Cloning shDAF16.1 in pYX232 vector
shDaf16Rev1	5′AGCTTAAAAAGGATACTATGGATGT TGATTCTCTTGAAATCAACATCCATAGTATCCGGG3′	
shDaf16For2	5′AATTCCCGGAGGGCAGCATATTAT TTTCAAGAGAAATGAATATGCTGCCCTCCTTTTTA3′	Cloning shDAF16.2 in pYX232 vector
shDaf16Rev2	5′AGCTTAAAAAGGGAGGGCAGCATATTCATTTCTCTTGAAAATAATATGCTGCCCTCCGGG3′	
sh232insFW	5′GTTAACGGTTTAGTGTTTTCTTACCCAATTGTAGAGACTAGGCAAGAGAGAAGACCCAGAGATG3′	Donor amplification
ACR3GFPinsREV	5′GAGTCCGGAACTCTAGCAGGCGCATAACATAAGTCACAAAGACGTTGTAAAACGACGGCC3′	
SeqshRNATPIFor	5′GGCTGCTGTAACAGGGAATATAAAGG3′	Sequencing to verify the integration of shRNA cassette
Daf16.1_PriEx	5′GGATACTATGGATGTTGATTCTCTTGAA3′	Labeled primers for primer extension
Daf16.2_PriEx	5′GGAGGGCAGCATATTCATTTCTCTTGAA3′	
PRIMER *daf-16* FOR	5′TCAAGACCTCAAAGCCAATCAACTC3′	Selective primers for *daf-16* gene used in the RT-qPCR analysis.
PRIMER *daf-16* REV	5′ACGAGAAAGAAGGAGTAAGAGGAG-3′	
PRIMER *cdc-42* FOR	5′CTGCTGGACAGGAAGATTACG3′	Selective primers for *cdc-42* gene used in the RT-qPCR analysis.
PRIMER *cdc-42* REV	5′CTCGGACATTCTCGAATGAAG3′	

## Data Availability

Data is contained within the article and Appendix A.

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
