# Peer review of "Yeast Oral Delivery of DAF16 shRNAs Results in Effective Gene Silencing in C. elegans"

_cimb, 2025, doi:10.3390/cimb47070570_

Round 1

Reviewer 1 Report

Comments and Suggestions for Authors

The authors describe construction of yeast strains expressing shRNA targeting the daf-16 gene in C. elegans. Using both yeast transformation and chromosomal integration resulted in efficient knockdown of the target gene, though the later was found to be more stable and efficient. Different shRNA structures also exhibited different efficiencies. These results demonstrate an efficient methodology for using yeast expression systems to target plant parasitic nematodes. The manuscript is well written, clear and easy to follow.

Minor comments:

Line 73-75: This sentence seems too long and complicated. Making it simpler will make it easy to grasp.

Line 93: Can RNAi be expressed or is dsRNA/shRNA etc. expressed?

Line 95: “.. resulted partially resistant to infection...” incorrect grammar  

Line 137: grown at 30°C vs growth

Line 256-258: I believe these were left here by mistake

Major comments:

There have been some studies, for example in certain lepidopteran insects, demonstrating that efficient knockdown doesn’t always translate to into mortality. Can this be true for C. elegans as well? Was the mortality tested in this study? Including that data is strongly recommended as it will greatly improve the importance and relevance of this manuscript.

Comments on the Quality of English Language

The overall writing and quality of English are quite good; however, there are some formatting and grammatical issues that need to be addressed.

Author Response

Line 73-75: This sentence seems too long and complicated. Making it simpler will make it easy to grasp.

Thanks for the suggestion. We rewrite this sentence in a clearer form

Line 93: Can RNAi be expressed or is dsRNA/shRNA etc. expressed?

In this paper the author expressed in plants hairpin-derived dsRNA targeting MiDaf16-like1 and MiSkn1-like1 genes individually (single-gene silencing) or simultaneously. We modified the text making it clearer

Line 95: “.. resulted partially resistant to infection...” incorrect grammar

Thanks for the suggestion. We corrected these mistakes

Line 137: grown at 30°C vs growth

We modified the text accordingly

Line 256-258: I believe these were left here by mistake

We thank the reviewer, actually it was left by mistake, we removed this text.

Major comments:

There have been some studies, for example in certain lepidopteran insects, demonstrating that efficient knockdown doesn’t always translate to into mortality. Can this be true for C. elegans as well? Was the mortality tested in this study? Including that data is strongly recommended as it will greatly improve the importance and relevance of this manuscript.

We appreciate the Reviewer’s comment. The selected target, daf-16, is indeed not an essential gene. Our goal was to assess the specificity and efficacy of gene targeting without inducing general toxicity or lethality. In line with the Reviewer’s suggestion, we performed a lifespan analysis on nematodes fed with either shDaf-16.∅ or shDaf-16.2, which was identified as the most effective construct. The results revealed a significant reduction in the survival rate of animals treated with shDaf-16.2 compared to those fed with shDaf-16.∅. Specifically, by day 11 of adulthood, the survival rate of the shDaf-16.2 group was 55%, whereas the control group showed an 80% survival rate (Figure 7 in the revised manuscript).

Reviewer 2 Report

Comments and Suggestions for Authors

The manuscript describes an interesting and useful phenomenon.  Several comments to improve:

  1. The  introduction should be revised.  There are several very long, dense paragraphs, followed by several which are only 1 sentence.  
  2. The major concern with the paper is in the interpretation of the results.  The shRNAs used here CLEARLY reduce expression of the target genes.  However, the authors did not include control experiments to demonstrate this is specific.  the empty vector controls are good, but the authors need to include EITHER a scramble sequence which would not bind to the target (and presumably not form a hairpin), and ideally a second control where an shRNA is used to target a different gene product.  These two controls will demonstrate that the shDaf16.1 and  shDaf16.2 are not only effective but SPECIFIC for their activity.
  3. The authors should discuss their choice of time point.  while further experiments in this are are beyond the scope of this paper, a discussion of WHY 1 day and how this may change with alternative time points is valuable.  Half life stability of the shRNAs should also be commented on regarding this.

Author Response

The introduction should be revised. There are several very long, dense paragraphs, followed by several which are only 1 sentence.

Thanks for this suggestion. We revised the introduction making it more fluent

The major concern with the paper is in the interpretation of the results.  The shRNAs used here CLEARLY reduce expression of the target genes. However, the authors did not include control experiments to demonstrate this is specific. the empty vector controls are good, but the authors need to include EITHER a scramble sequence which would not bind to the target (and presumably not form a hairpin), and ideally a second control where an shRNA is used to target a different gene product.

These two controls will demonstrate that the shDaf16.1 and  shDaf16.2 are not only effective but SPECIFIC for their activity.

We thank the reviewer for these observations. Actually, the possibility to have off target is a potential threat and for this reason we repeated the experiment including the unrelated gene cdc-42. As shown in a new supplementary figure S2, there is no effect on the expression of this genes both with the empty or with the shDaf16.2 constructions, showing the specificity of this construct in lowering DAF16 mRNA.

The authors should discuss their choice of time point.  while further experiments in this are are beyond the scope of this paper, a discussion of WHY 1 day and how this may change with alternative time points is valuable.  Half life stability of the shRNAs should also be commented on regarding this.

We thank the Reviewer for this comment. We chose to analyze the phenotype at day 1 of adulthood to ensure that we captured the direct and specific effects of the gene inactivation, minimizing the potential influence of compensatory mechanisms that could be activated at later day of adulthood. However, a lifespan analysis of nematodes fed with shDaf-16.∅ or shDaf-16.2 was performed (Figure 7 in the revised version), highlighting a significant reduction in the survival rate of shDaf-16.2- treated animals, with respect to shDaf-16.∅-fed nematodes. Concerning the shRNA stability, we always used for worm feeding the same amount of yeast cells from exponential phase, assuming that there is no effect of shRNA stability or, if any, it does not seem to interfere with the results. Anyway, this is a good point to study with the aim to improve the efficacy of the system, but it is out of the scope of this paper.

Round 2

Reviewer 2 Report

Comments and Suggestions for Authors

much improved